# OpenReview forum: "User-Assistant Bias in LLMs"
_ICLR.cc/2026/Conference — ICLR 2026 Conference Withdrawn Submission_

### Official Review · Reviewer_ztJz · 2025-10-17

**Soundness:** 4
**Presentation:** 3
**Contribution:** 3
**Rating:** 4
**Confidence:** 3

**Summary:**

This paper introduces UserAssist, a synthetic dataset designed to benchmark user-assistant bias in multi-turn conversations. The authors identify two key phenomena: user bias and assistant bias in conversations, showing how these biases affect conversational dynamics. The paper proposes a new metric, user-assistant bias, which measures the tendency of models to align more with the user or assistant. Through fine-tuning on this dataset, the authors demonstrate that preference alignment and reasoning-heavy training can mitigate biases in LLMs. The paper also shows that user-assistant bias can be effectively manipulated through direct preference optimization (DPO), leading to models that balance both user and assistant perspectives.

**Strengths:**

1. The introduction of USERASSIST as a dataset to measure user-assistant bias is novel and insightful. This dataset provides a direct and objective way to evaluate biases in conversational models, addressing an underexplored aspect of LLM performance.
2. The paper’s metrics, such as user-assistant bias and logprob, are well-thought-out and offer a meaningful approach to evaluating how well models balance user and assistant perspectives. These metrics are intuitive and make sense.

**Weaknesses:**

1. Real-World Data Performance: In Figure 5, the performance on in-distribution data is clear and effective, with the metrics trending toward ±1. However, the performance on real-world datasets in Figure 6 is more limited. The authors should discuss potential reasons for this discrepancy. Since the direction of preference optimization should ideally be consistent, regardless of the task, why does the real-world data show weaker performance?
2. Possible Extensions of UserAssist: The paper would benefit from a discussion on the potential for constructing the USERASSIST dataset using real-world multi-turn dialogues, such as those found in Wildchat. Would the dataset benefit from being based on real-world conversations to capture more authentic user-assistant dynamics?
3. Model Preference Learning: The paper suggests that user-assistant bias can be bidirectionally adjusted. However, it would be interesting to explore the underlying reasons behind this. Is it possible that models are learning to prioritize special tokens like `<|im_start|>user` and `<|im_start|>assistant`, rather than truly learning to favor the user or assistant? This is something worth investigating further to ensure that the bias adjustment is meaningful and not just a result of token-based patterns.

**Questions:**

In Figure 6, the paper shows that larger models tend to have a higher initial user bias. However, the effect of training on USERASSIST to control this bias seems independent of model size. Could the authors provide a clearer explanation for this observation? Why does model size not affect the magnitude of the bias adjustment?

---

### Official Review · Reviewer_7zta · 2025-10-27

**Soundness:** 2
**Presentation:** 3
**Contribution:** 2
**Rating:** 4
**Confidence:** 4

**Summary:**

This paper investigates user–assistant bias — a previously underexplored form of bias emerging in large language model (LLM) interactions, where the model’s output is systematically influenced by the perceived identity or stance of the user. Unlike traditional bias research focusing on model outputs in isolation, this work examines interactive dynamics between users and assistants.

**Strengths:**

1.    **Conceptual Innovation** The paper introduces the novel concept of user–assistant bias, expanding the scope of bias research from static text generation to interactive dialogue contexts. This reframing moves beyond conventional fairness evaluations centered on stereotypical content, highlighting how user framing itself can influence model behavior—an underexplored yet crucial dimension of alignment.

2.    **Comprehensive and Cross-Model Evaluation** The study benchmarks over 50 leading LLMs, including both commercial and open-weight families, covering reasoning, chat, and distilled models. This breadth allows for systematic cross-model comparison, enhancing the generalizability and robustness of the conclusions.

3.    **Broader Significance** The work underscores the need to evaluate fairness in interactive and dynamic contexts, rather than static single-turn tasks.

**Weaknesses:**

1.    **Critical Analysis of the USERASSIST Design.** The dialogues in USERASSIST are highly symbolic and detached from real-world semantics (e.g., “A=20” vs. “A=70”). While this design facilitates experimental control and quantitative measurement of bias, it drastically oversimplifies the complexity of authentic human–AI interactions. In real-world scenarios, conflicts between users and assistants typically arise from richer cognitive and social contexts—for example, users making requests that violate ethical or safety guidelines, spreading misinformation or factual misunderstandings, models generating hallucinations, ambiguous or implicit stance expressions, or the model exhibiting “sycophantic” behavior under social pressure. In contrast, USERASSIST captures merely formalized conflicts, rather than semantic or socially grounded ones.

---

2.    **A model’s responses in such abstract settings are more likely to reflect language distributional preferences rather than genuine role-based preferences.** The model may not even recognize who the “user” or “assistant” is; thus, alignment with the user’s answer cannot be straightforwardly interpreted as “the model trying to please the user.” Such alignment might instead result from prior linguistic probabilities, recency bias, co-occurrence frequencies, or decoder sampling tendencies.

---

3.    **The definition of “user–assistant bias” itself carries conceptual ambiguity.** The metric implicitly assumes that if a model more frequently agrees with the user, it exhibits greater deference or obedience. However, “obedience” or “sycophancy” is inherently a social–psychological behavior that requires semantic grounding and social context—something that symbolic assignments alone cannot meaningfully capture.

**Questions:**

see Weaknesses

---

### Official Review · Reviewer_3YSp · 2025-10-31

**Soundness:** 1
**Presentation:** 3
**Contribution:** 1
**Rating:** 2
**Confidence:** 4

**Summary:**

This paper constructs a new dataset to evaluate to what extent LLM's responses bias toward past user or AI messages. Extensive experiments are conducted based on commericial and open source models. The paper also studies a wide range of finetuning techniques and how they mitigate/exacerbate the bias problem.

**Strengths:**

* Overly relying on previous user or AI messages is an important problem, and this paper presents a reasonable first attempt.

* The evaluation is done extensively, accounting for both commercial and open-source LLMs.

* The finetuning experiments are well done and well presented, including testing on more real human-AI conversations.

**Weaknesses:**

* The problem of User-Assistant bias is ill-defined. The real problem is not about LLM refers to AI or user, but about whether it refers to truth, right answer, or what leads to good outcome of a conversation. The symbol-value and object-color constructs are overly simplified and does not have much practical real world value. Take examples that authors show in Figure 1 as an example, there is not necessarily a problem if LLM takes the value of x from user or assistant as there is no good/bad judgement. Is LLM biased towards user good in this case? It doesn't seem to be clear.

* The same problem applies to synthetic debate dataset (not really realistic as it's purely synthesized by LLM). The evaluation task isn't well defined either. The LLMs trained using these datasets seem to blindly bias towards user or assistant. What we really need the model to do is to intelligently reason what is right, what is best to the user, what is best for the session/context. That type of "debiasing" will have more meaningful.

**Questions:**

Please see points in weaknesses

---

### Official Review · Reviewer_VLSf · 2025-11-02

**Soundness:** 3
**Presentation:** 2
**Contribution:** 3
**Rating:** 4
**Confidence:** 2

**Summary:**

### **Summary**

This paper introduces and formalizes the concept of **“user–assistant bias”** in Large Language Models (LLMs). This bias arises from the presence of **conflicting information** in multi-turn conversations, causing LLMs to become either overly **agreeable** (favoring the user’s input) or overly **stubborn** (favoring their own prior outputs).

To study this, the authors create **USERASSIST**, a novel benchmark consisting of about **8,000 synthetic multi-turn dialogues** in which user and assistant roles provide **equal, conflicting, and stance-neutral information** in a balanced, task-agnostic manner. Using this dataset, they evaluate **52 models** (26 commercial and 26 open-weight) and find a clear trend: **base and reasoning-tuned models remain close to neutral**, while **instruction-tuned models** (e.g., GPT-4o) exhibit **strong user bias**. In contrast, models such as **Claude 3.7 Sonnet, Claude 4 Sonnet, o1-preview, o4-mini, DeepSeek Reasoner, Gemini 2.5 Flash Preview, and Grok 3 Mini** show minimal bias toward either side.

**Strengths:**

### **Strengths of the Experiments**

1.  **The authors create USERASSIST**.
      A novel benchmark consisting of about **8,000 synthetic multi-turn dialogues** in which user and assistant roles provide **equal, conflicting, and stance-neutral information** in a balanced, task-agnostic manner.

1. **Comprehensive model coverage.**
   The authors evaluate 52 models in total, including 26 commercial and 26 open-weight models. This large coverage gives the study impressive scope and helps reveal consistent patterns across major model families such as GPT, Claude, Llama, and Qwen.

2. **Systematic fine-tuning studies.**
   The paper includes carefully controlled fine-tuning experiments using DPO and SFT to examine whether the bias can shift toward the user or assistant side. The results show clear directional effects: human-preference alignment through DPO increases user bias, while reasoning-trace fine-tuning through SFT reduces user bias.

**Weaknesses:**

### **Weaknesses**

1. **Limited realism of the setup.**
   USERASSIST-TEST provides stance-neutral, symmetric information — both user and assistant offer conflicting but equally simple facts (e.g., colors or numbers). While this design removes topic bias, it also oversimplifies real dialogues and may not capture complex social dynamics behind user–assistant interactions.


2. **Influence of in-context similarity.**
   The model may align more strongly when previous conversation snippets are textually or semantically similar to the current query. When they differ, the model seems more “stubborn.” This effect reflects **context similarity**, not necessarily a role-based bias—something the paper does not control or analyze.


### Overall assessment:

The dataset design is conceptually interesting and theoretically clean; however, the paper’s validation is not rigorous enough to confirm that the observed “user–assistant bias” reflects a genuine role-conditioned behavior rather than artifacts such as position, recency, or similarity effects. USERASSIST-TEST provides the model with uniform, stance-neutral information, so any leaning toward the user (sycophancy) or toward the assistant (stubbornness) may stem from subtle context-level cues rather than true preference. In this setup, prior conversation turns act merely as in-context examples, if they closely resemble the current query, the model’s answer naturally aligns with them; if not, it appears more resistant. Without additional control (e.g., similarity-conditioned analysis,), it is difficult to conclude that the measured differences are caused by “bias” rather than by contextual similarity or positional influence. Overall, the experiments are extensive and systematic, but the causal interpretation remains speculative and would benefit from stronger validation and ablation evidence.

**Questions:**

### **Questions and Suggestions for the Authors**

1. **Sampling temperature mismatch**

   * Closed-source models were evaluated at *temperature = 1* (fixed by the API), whereas open-weight models were tested deterministically at *temperature = 0*.
   * Could the authors re-evaluate open-weight models at *temperature = 1* (or with comparable sampling variance) to confirm that observed bias differences are not merely artifacts of decoding randomness?
   * A brief analysis of sensitivity to sampling temperature would clarify the robustness of cross-model comparisons.

2. **Influence of in-context similarity**

   * Prior conversation turns may act as in-context exemplars; if their lexical or semantic structure resembles the query, the model might preferentially reproduce those segments regardless of role identity.
   * Could the authors measure **cosine similarity between the query and preceding turns**, and report bias conditional on similarity bins (e.g., high- vs. low-similarity contexts)?
   * Such an analysis would help distinguish *true role bias* from *contextual similarity effects*.

3. **Possible misinterpretation of CoT effects**

   * The paper claims that reasoning-style (Chain-of-Thought) fine-tuning mitigates user–assistant bias by promoting balanced information use.
   * However, the effect might instead arise from an **in-context sampling advantage**: reasoning models tend to retrieve and integrate broader context more effectively, reducing sensitivity to superficial recency or similarity cues.
   * Could the authors verify this by inspecting **context-similarity patterns** before and after CoT fine-tuning?

---

### Note · Authors · 2025-11-21

I have read and agree with the venue's withdrawal policy on behalf of myself and my co-authors.